# The effect of menopause on cardiovascular risk factors according to body mass index in middle-aged Korean women

Do Kyeong Song, Young Sun Hong, Yeon-Ah Sung, Hyejin Lee *

Department of Internal Medicine, Ewha Womans University School of Medicine, Seoul, Korea

* hyejinlee@ewha.ac.kr

**Data Availability Statement:** All relevant data are within the paper.

**Funding:** The author(s) received no specific funding for this work.

## Abstract

### Background

Menopausal status and obesity are associated with an increased risk for cardiovascular diseases. However, there are few studies on the effect of menopause on cardiovascular risk factors according to the degree of obesity during the menopausal transition. We aimed to evaluate the effect of menopause on cardiovascular risk factors according to body mass index (BMI) in middle-aged Korean women.

### Methods

We analyzed 361 postmenopausal women and 758 premenopausal women (age: 45–55 years) without diabetes mellitus, hypertension, or dyslipidemia, using a cohort database released by the Korean National Health and Nutrition Examination Survey 2016–2018. Subjects were divided into two groups based on BMI. Women who underwent a hysterectomy or were pregnant were excluded from this study. Differences between groups adjusted for age and BMI were assessed.

### Results

Postmenopausal women (52 ± 2 years) were older than premenopausal women (48 ± 2 years), and BMI did not differ between the two groups (22.8 ± 2.9 vs. 23.0 ± 3.1 kg/m$^2$). After adjustment for age and BMI in total and non-obese subjects (not obese subjects), postmenopausal women exhibited higher hemoglobin A1c and total cholesterol levels than premenopausal women. Subgroup analysis for 138 postmenopausal and 138 age- and BMI-matched premenopausal women showed that postmenopausal women had higher total cholesterol levels than premenopausal women with marginal significance (201 ± 25 vs. 196 ± 27 mg/dL).

### Conclusion

Menopausal status was associated with increased glucose and cholesterol levels independent of age and BMI in middle-aged Korean women. Menopausal status showed a significant relationship with increased total cholesterol levels even after adjusting for age and BMI

**Competing interests:** The authors have declared that no competing interests exist.

in non-obese women but not obese women. Therefore, intensive monitoring and treating of lipid status is necessary to prevent cardiovascular events during the menopausal transition, especially in non-obese subjects.

## Introduction

Cardiovascular disease is the leading cause of death for women [1]. Menopausal status is associated with an increased risk for cardiovascular diseases mainly due to changes in body fat distribution, glucose metabolism, and serum lipids [2]. Traditionally, menopause is defined as the absence of menstruation with no other cause for 12 consecutive months [3]. The menopausal transition is characterized by ovarian hormone changes, menstrual cycle irregularities, and an increased risk for cardiovascular diseases [4]. Cardiovascular disease incidence rates were higher in postmenopausal women than premenopausal women in each age group among women below 55 years in a cohort of 2873 Framingham women [5]. A rise in coronary heart disease incidence after menopause was also noted in the Framingham study [6].

Previous studies on changes in cardiovascular risk factors among women during the menopausal transition demonstrate inconsistent results. Most epidemiological studies show that changes in lipid metabolism during the menopausal transition are associated with a more atherogenic lipid profile, and menopause is associated with increased total cholesterol and low-density lipoprotein (LDL) cholesterol levels. However, there are variations in the lipid profiles during the menopausal transition between studies [7–10]. A cross-sectional study conducted on Korean women (age: 44–56 years) showed that blood pressure was significantly higher during late (than early) menopausal transition [11]. However, blood pressure depended more on age than menopausal status among middle-aged Korean women after excluding women on medication for hypertension [8]. It is debatable whether menopause increases cardiovascular risk factors independent of aging. Although the risk of cardiovascular disease linearly increased with increased body mass index (BMI) in premenopausal and postmenopausal Korean women, using a nationwide health examination database [12], the effect of BMI on cardiovascular risk factors during the menopausal transition among women in Korea was unclear.

Furthermore, the effect of postmenopausal status on cardiovascular risk factors is known to vary by ethnicity, partly due to different lifestyles and dietary patterns. Ethnic differences in serum reproductive hormone concentrations exist independent of menopausal status. Studies show that Chinese and Japanese women exhibit lower estradiol concentrations during the menopausal transition than Caucasian women [13]. To date, there are few studies on the effect of menopause on cardiovascular risk factors according to the degree of obesity during the menopausal transition among women in Korea. We aimed to evaluate the effect of menopause on cardiovascular risk factors according to BMI in middle-aged Korean women.

## Materials and methods

### Data source

We used the data from the Korea National Health and Nutrition Examination Survey (KNHANES) from 2016 to 2018. The KNHANES is a national surveillance system assessing the health and nutritional status of non-institutionalized Korean citizens since 1998. This nationally representative cross-sectional survey collects data on socioeconomic status, health-related behaviors, quality of life, healthcare utilization, dietary intake, anthropometric measures, and biochemical profiles using fasting blood serum and urine and clinical profiles for

major chronic diseases through a health interview, nutrition survey, and annual health examinations. The health interviews and physical health examinations are conducted by trained medical staff and interviewers [14]. Because KNHANES data comprise nationally representative samples of Korea, including the health interview, physical examination, and nutrition survey, they are valuable resources for evaluating the relationship between risk factors and diseases in Korea.

We did not obtain informed consent from individuals because we did not collect data for the study. The patient records were anonymous before being released by the KNHANES. This study was approved by the Institutional Review Board of Ewha Medical Center. All methods followed the relevant guidelines and regulations.

### Study population & outcome variables

We included women (age: 45–55 years) using a cohort database released by the KNHANES 2016–2018. The study excluded women who underwent a hysterectomy or were pregnant and subjects diagnosed with diabetes, hypertension, or dyslipidemia (based on self-reported questionnaires). Finally, we enrolled 361 postmenopausal and 758 premenopausal women 45–55 years.

The BMI was calculated as body weight in kilograms divided by height in meters squared. Body weight and height were measured during the health examinations. Subjects were divided into two groups according to BMI (non-obese subjects: BMI $< 25$ kg/m$^2$, obese subjects: BMI $\geq 25$ kg/m$^2$), following Asian-specific criteria [15]. Menopausal status was determined based on self-reported questionnaires; we categorized menopause status as pre- and postmenopausal. Subgroup analysis included 138 postmenopausal women and 138 age- and BMI-matched premenopausal women.

Blood pressure was calculated as the mean of two manual sphygmomanometer readings with patients in sitting positions. A blood sample was obtained in the morning after an overnight fast. Total cholesterol, high-density lipoprotein (HDL) cholesterol, triglycerides, fasting glucose, and hemoglobin A1c (HbA1c) were measured, and LDL cholesterol was calculated using the Friedewald equation [16].

### Statistical analysis

The Kolmogorov-Smirnov statistic was used to analyze the continuous variables for normality. Quantitative variables were reported as the means ± the standard deviations. The between-group differences were assessed using the Student unpaired t-tests. Differences between groups adjusted for age and BMI were assessed using analysis of covariance. Multiple linear regression analyses were performed to determine the independent association between menopausal status and total cholesterol after controlling for age, BMI, systolic blood pressure, and fasting glucose and confirm the association between menopausal status and HbA1c after controlling for age, BMI, systolic blood pressure, and total cholesterol. Statistical analysis was performed using the SPSS 23.0 software package for Windows (IBM Corporation, Chicago, IL, USA). *P* values $< 0.05$ were considered statistically significant.

### Results

Table 1 represents the baseline characteristics of participants by menopausal status. Postmenopausal women were older than premenopausal women ($P < 0.05$). The mean age was 48 years for premenopausal women and 52 years for postmenopausal women. BMI did not differ between the two groups. After adjustment for age and BMI, postmenopausal women had higher hemoglobin A1c (HbA1c) and total cholesterol levels than premenopausal women. After adjustment for age and BMI, fasting plasma glucose, HDL cholesterol, triglycerides, LDL

**Table 1. Baseline characteristics of study participants.**

| | Total subjects | | | |
| | Premenopausal women (n = 758) | Postmenopausal women (n = 361) | P-value | Adjusted P-value* |
|---|---|---|---|---|
| Age (y) | 48 ± 2 | 52 ± 2 | <0.001 | |
| BMI (kg/m²) | 23.0 ± 3.1 | 22.8 ± 2.9 | 0.395 | |
| Fasting glucose (mg/dL) | 93 ± 12 | 94 ± 9 | 0.241 | 0.122 |
| HbA1c (%) | 5.4 ± 0.4 | 5.5 ± 0.3 | <0.001 | 0.002 |
| Total cholesterol (mg/dL) | 191 ± 28 | 201 ± 25 | <0.001 | 0.045 |
| HDL cholesterol (mg/dL) | 55 ± 12 | 56 ± 13 | 0.366 | 0.650 |
| Triglycerides (mg/dL) | 96 ± 56 | 101 ± 48 | 0.190 | 0.499 |
| LDL cholesterol (mg/dL) | 116 ± 25 | 125 ± 22 | <0.001 | 0.087 |
| Systolic BP (mmHg) | 110 ± 11 | 110 ± 12 | 0.209 | 0.185 |
| Diastolic BP (mmHg) | 73 ± 8 | 73 ± 8 | 0.214 | 0.977 |

Plus-minus values are the means ± the standard deviations.

* Adjusted for age and body mass index.

BMI, body mass index; HbA1c, hemoglobin A1c; HDL cholesterol, high-density lipoprotein cholesterol; LDL cholesterol, low-density lipoprotein cholesterol; BP, blood pressure.

cholesterol, systolic blood pressure, and diastolic blood pressure did not differ between post-menopausal and premenopausal women (Table 1).

On dividing the subjects into two groups according to BMI, 33.6% (n = 291) women were postmenopausal in non-obese subjects, and 27.7% (n = 70) women were postmenopausal in obese subjects. Postmenopausal women were older than premenopausal women in non-obese and obese groups (all Ps < 0.05). After adjustment for age and BMI, postmenopausal women exhibited higher HbA1c and total cholesterol levels than premenopausal women only in non-obese subjects. Fasting glucose, HDL cholesterol, triglycerides, LDL cholesterol, systolic blood pressure, and diastolic blood pressure did not differ between postmenopausal and premeno-pausal women in non-obese and obese subjects after adjustment for age and BMI (Table 2).

The association between menopausal status and total cholesterol according to BMI was similar in the subgroup analysis. Subgroup analysis for 138 postmenopausal and 138 age- and BMI-matched premenopausal women showed that postmenopausal women had higher total cholesterol levels than premenopausal women with marginal significance in total (201 ± 25 mg/dL vs. 196 ± 27 mg/dL, P = 0.091) and non-obese subjects (201 ± 25 mg/dL vs. 194 ± 25 mg/dL, P = 0.060); however, total cholesterol levels did not differ between postmenopausal and premenopausal women in obese subjects (206 ± 23 mg/dL 141 vs. 205 ± 36 mg/dL, P = 0.944). The levels of fasting glucose, HbA1c, HDL cholesterol, triglycerides, LDL cholesterol, systolic blood pressure, and diastolic blood pressure did not differ between postmeno-pausal and age- and BMI- matched premenopausal women (Table 3).

Multiple linear regression analyses showed that menopausal status was an independent determinant of total cholesterol after adjustment for age, BMI, systolic blood pressure, and fasting glucose (β = 4.58, P < 0.05) in the total number of subjects (Table 4). Menopausal status was also an independent determinant of HbA1c after adjustment for age, BMI, systolic blood pressure, and total cholesterol (β = 0.08, P < 0.01) in the total number of subjects (Table 5).

## Discussion

Using data from the KNHANES from 2016 to 2018, we demonstrated that menopausal status was associated with increased glucose and cholesterol levels, independent of age and BMI,

**Table 2. Comparison of characteristics and cardiovascular risk factors in study participants according to the degree of obesity.**

| | Non-obese subjects | | | | Obese subjects | | | |
|---|---|---|---|---|---|---|---|---|
| | Premenopausal women (n = 575) | Postmenopausal women (n = 291) | P-value | Adjusted P-value* | Premenopausal women (n = 183) | Postmenopausal women (n = 70) | P-value | Adjusted P-value* |
| Age (y) | 48 ± 3 | 52 ± 2 | <0.001 | | 48 ± 2 | 53 ± 2 | <0.001 | |
| BMI (kg/m$^2$) | 21.6 ± 1.9 | 21.7 ± 1.8 | 0.322 | | 27.3 ± 2.1 | 27.3 ± 2.2 | 0.996 | |
| Fasting glucose (mg/dL) | 92 ± 8 | 93 ± 8 | 0.013 | 0.091 | 97 ± 18 | 97 ± 11 | 0.776 | 0.580 |
| HbA1c (%) | 5.4 ± 0.3 | 5.5 ± 0.3 | <0.001 | 0.001 | 5.5 ± 0.5 | 5.6 ± 0.3 | 0.512 | 0.483 |
| Total cholesterol (mg/dL) | 190 ± 27 | 201 ± 25 | <0.001 | 0.032 | 194 ± 28 | 201 ± 24 | 0.045 | 0.966 |
| HDL cholesterol (mg/dL) | 57 ± 12 | 58 ± 12 | 0.506 | 0.185 | 51 ± 11 | 50 ± 11 | 0.848 | 0.064 |
| Triglycerides (mg/dL) | 90 ± 42 | 95 ± 42 | 0.066 | 0.549 | 117 ± 82 | 123 ± 64 | 0.584 | 0.669 |
| LDL cholesterol (mg/dL) | 115 ± 24 | 124 ± 22 | <0.001 | 0.123 | 120 ± 26 | 126 ± 23 | 0.061 | 0.575 |
| Systolic BP (mmHg) | 108 ± 11 | 110 ± 12 | 0.155 | 0.258 | 113 ± 11 | 114 ± 13 | 0.481 | 0.547 |
| Diastolic BP (mmHg) | 72 ± 7 | 73 ± 7 | 0.210 | 0.989 | 75 ± 7 | 76 ± 8 | 0.369 | 0.855 |

Plus-minus values are the means ± the standard deviations.

* Adjusted for age and body mass index.

BMI, body mass index; HbA1c, hemoglobin A1c; HDL cholesterol, high-density lipoprotein cholesterol; LDL cholesterol, low-density lipoprotein cholesterol; BP, blood pressure.

among middle-aged Korean women. Multiple linear regression analyses showed that menopausal status was an independent determinant of total cholesterol and HbA1c. Menopausal status showed a significant relationship with increased total cholesterol levels, even after adjusting for age and BMI in non-obese women. However, total cholesterol levels did not differ between postmenopausal and premenopausal subjects in obese women.

The effects of menopause on the lipid profiles during the menopausal transition in our study were comparable with the results of previous studies. Numerous epidemiological studies suggest menopause-associated changes in the lipid profile. In the Study of Women's Health Across the Nation (a longitudinal, community-based, multiethnic population study), total cholesterol and LDL cholesterol levels increased significantly within a year of the final menstrual period in 1,054 women who achieved the final menstrual period by the end of 9 years of follow-up during the menopausal transition [10]. In middle-aged Caucasian women (age: 47–55 years), the menopausal transition was associated with increased total cholesterol, LDL cholesterol, and HDL cholesterol levels, independent of age [17]. In 593 healthy Chinese women 35 to 64 years, late perimenopausal status showed a significant association with an accelerated increase in total cholesterol and triglycerides. However, HDL cholesterol levels did not differ among different menopausal status groups [9]. Of the 1,169 perimenopausal Korean women (age: 40–64 years) from the KNHANES 2005, postmenopausal women exhibited higher total cholesterol and LDL cholesterol levels than premenopausal women. However, there were no significant differences in HDL cholesterol levels between premenopausal and postmenopausal women after excluding subjects on medications for hypercholesterolemia [8]. Consistent with the results of our study, total cholesterol level was higher in postmenopausal women than premenopausal women, mostly in previous studies. Although we could not clarify the mechanism

**Table 3. Comparison of characteristics and cardiovascular risk factors in age- and BMI-matched participants.**

| | Total subjects | | | Non-obese subjects | | | Obese subjects | | |
|---|---|---|---|---|---|---|---|---|---|
| | Premenopausal women (n = 138) | Postmenopausal women (n = 138) | P-value | Premenopausal women (n = 117) | Postmenopausal women (n = 117) | P-value | Premenopausal women (n = 21) | Postmenopausal women (n = 21) | P-value |
| Age (y) | 51 ± 2 | 51 ± 2 | 1.000 | 51 ± 2 | 51 ± 2 | 1.000 | 51 ± 2 | 51 ± 2 | 1.000 |
| BMI (kg/m$^2$) | 22.5 ± 2.4 | 22.6 ± 2.4 | 0.792 | 21.7 ± 1.6 | 21.8 ± 1.6 | 0.653 | 26.7 ± 1.8 | 26.7 ± 1.7 | 0.960 |
| Fasting glucose (mg/dL) | 93 ± 9 | 93 ± 7 | 0.710 | 93 ± 9 | 93 ± 7 | 0.836 | 95 ± 8 | 96 ± 7 | 0.602 |
| HbA1c (%) | 5.5 ± 0.3 | 5.5 ± 0.3 | 0.099 | 5.5 ± 0.4 | 5.5 ± 0.3 | 0.152 | 5.4 ± 0.3 | 5.5 ± 0.3 | 0.381 |
| Total cholesterol (mg/dL) | 196 ± 27 | 201 ± 25 | 0.091 | 194 ± 25 | 201 ± 25 | 0.060 | 205 ± 36 | 206 ± 23 | 0.944 |
| HDL cholesterol (mg/dL) | 56 ± 11 | 59 ± 13 | 0.116 | 56 ± 11 | 59 ± 14 | 0.051 | 56 ± 9 | 53 ± 10 | 0.313 |
| Triglycerides (mg/dL) | 93 ± 39 | 99 ± 46 | 0.246 | 90 ± 37 | 96 ± 44 | 0.290 | 109 ± 43 | 118 ± 50 | 0.575 |
| LDL cholesterol (mg/dL) | 122 ± 23 | 123 ± 21 | 0.497 | 120 ± 22 | 122 ± 21 | 0.508 | 127 ± 31 | 129 ± 21 | 0.815 |
| Systolic BP (mmHg) | 110 ± 11 | 108 ± 12 | 0.387 | 109 ± 11 | 109 ± 12 | 0.808 | 114 ± 11 | 108 ± 12 | 0.106 |
| Diastolic BP (mmHg) | 73 ± 7 | 73 ± 8 | 0.527 | 72 ± 7 | 73 ± 7 | 0.318 | 76 ± 5 | 74 ± 9 | 0.490 |

Plus-minus values are the means ± the standard deviations.

BMI, body mass index; HbA1c, hemoglobin A1c; HDL cholesterol, high-density lipoprotein cholesterol; LDL cholesterol, low-density lipoprotein cholesterol; BP, blood pressure.

of change in lipid profiles during the menopausal transition in this study, decreasing estrogen levels during the menopausal transition known to affect hepatic lipase and lipoprotein lipase activity [18–20] may have an important role in the lipid metabolism during the menopausal transition. However, HDL and LDL cholesterol results according to the menopausal status were inconsistent between studies. We estimated LDL cholesterol using the Friedewald equation, contrary to the study on Caucasian women [17]. The age range of the participants was narrow in our study compared to previous studies on Chinese [9] or Korean women [8]. The age of the study participants, differences in methods measuring cholesterol levels, or different

**Table 4. Association of menopausal status with total cholesterol by multiple linear regression analyses.**

| Variable | Effect of menopausal status | | 95% CI | P-value |
|---|---|---|---|---|
| | β | SE | | |
| Menopausal status-unadjusted | 10.47 | 1.73 | 7.08–13.87 | <0.001 |
| BMI-adjusted | 10.59 | 1.72 | 7.21–13.97 | <0.001 |
| Age, BMI-adjusted | 4.48 | 2.23 | 0.11–8.85 | 0.045 |
| Age, BMI, SBP-adjusted | 4.59 | 2.23 | 0.21–8.96 | 0.040 |
| Age, BMI, SBP, fasting glucose-adjusted | 4.58 | 2.23 | 0.20–8.96 | 0.041 |

SE, standard error; 95% CI, 95% confidence interval; BMI, body mass index; SBP, systolic blood pressure.

**Table 5. Association of menopausal status with hemoglobin A1c by multiple linear regression analyses.**

| Variable | Effect of menopausal status | | 95% CI | *P*-value |
|---|---|---|---|---|
| | β | SE | | |
| Menopausal status-unadjusted | 0.12 | 0.02 | 0.08–0.17 | <0.001 |
| BMI-adjusted | 0.13 | 0.02 | 0.08–0.17 | <0.001 |
| Age, BMI-adjusted | 0.09 | 0.03 | 0.03–0.14 | 0.002 |
| Age, BMI, SBP-adjusted | 0.09 | 0.03 | 0.03–0.15 | 0.002 |
| Age, BMI, SBP, total cholesterol-adjusted | 0.08 | 0.03 | 0.03–0.14 | 0.004 |

SE, standard error; 95% CI, 95% confidence interval; BMI, body mass index; SBP, systolic blood pressure.

ethnicities may be responsible for the differences in LDL and HDL cholesterol results in women according to the menopausal status between studies.

The metabolic syndrome is characterized by abdominal adiposity, insulin resistance, and dyslipidemia and is closely associated with cardiovascular diseases [21]. The prevalence of metabolic syndrome increased during the menopausal transition in 949 women without diabetes or the metabolic syndrome at baseline was independent of aging in the Study of Women's Health Across the Nation [22]. Postmenopausal status was associated with an increased risk of metabolic syndrome after adjusting for age in 2,671 women who did not receive hormone replacement therapy in the KNHANES 2001 [23]. Postmenopausal status was an independent risk factor for metabolic syndrome after adjustment for age and BMI in 1,002 Korean women who participated in annual health examinations [24]. Postmenopausal status was associated with dysglycemia independent of aging in Japanese individuals [25]. A prospective study including 1,303 British women (age: 53 years) showed that HbA1c levels increased across the natural menopause transition after adjustment for BMI [26]. In a cross-sectional study, postmenopausal women showed higher HbA1c levels than premenopausal women after adjustments for age and BMI in Chinese women with BMI < 30 kg/m$^2$ [27]. HbA1c estimates long-term glucose status and predicts cardiovascular disease better than fasting or post-challenge glucose in women without diabetes mellitus [28]. Consistent with the results of previous studies, postmenopausal women had higher HbA1c levels than premenopausal women in our subjects after adjustment for age and BMI; menopausal status was an independent determinant of HbA1c in multiple linear regression analyses. Although we did not estimate abdominal obesity indicators, such as waist circumference or waist circumference/hip ratio, abdominal obesity accompanying menopause may be associated with increased insulin resistance and glucose levels in postmenopausal women [29].

In a study using the database from the KNHANES 2007–2010, waist circumference was significantly associated with systolic blood pressure after adjustment for age and BMI in 1,422 women (age: 45–55 years) during the menopausal transition. Waist circumference was associated with systolic blood pressure in non-obese women but not in obese women in this study group [30]. A cross-sectional study conducted in health-screening centers involving 2,037 Korean women (age: 44–56 years) showed significantly higher systolic blood pressure and diastolic blood pressure values during the late (than early) menopausal transition [11]. Blood pressure depended more on age than the menopausal status among middle-aged Korean women from the KNHANES 2005, after excluding women on medication for hypertension [8]. Although we did not evaluate the blood pressure between late and early menopausal transition or the association between waist circumference and blood pressure among women during the menopausal transition, blood pressure did not differ between postmenopausal and

premenopausal women after adjustment for age and BMI in our study, consistent with the result of the previous KNHANES 2005.

The effect of menopause on cardiovascular risk factors differed according to BMI among middle-aged Korean women in our study. Menopausal status was associated with increased total cholesterol levels only in non-obese women. Among 2,659 women followed in the Study of Women's Health Across the Nation annually for up to 7 years, both LDL cholesterol and total cholesterol peaked in late peri- and early menopause; however, increases in LDL cholesterol and total cholesterol were smallest in the highest baseline weight tertile [31]. A cross-sectional study involving 1,553 Korean women (age: 44–56 years) who underwent a health screening examination showed that an increased prevalence of high non-HDL cholesterol was associated with postmenopausal status and more pronounced in lean women than in overweight or obese women after excluding subjects on lipid-lowering medications or history of hypercholesterolemia [32]. Studies show that the effect of BMI on serum reproductive hormone levels varies by menopausal status. Increasing BMI was associated with decreasing estradiol levels in premenopausal women and increasing estradiol levels in postmenopausal women among 3,257 participants in the Study of Women's Health Across the Nation [13]. Relatively high estradiol levels in postmenopausal women with higher BMI may affect the cardiovascular risk factors differently when compared to non-obese postmenopausal women. Further studies measuring levels of reproductive hormones concurrently to evaluate the mechanism of the BMI effect on cardiovascular risk factors during the menopausal transition are needed.

This study is the first to evaluate the effect of menopause on cardiovascular risk factors according to BMI in middle-aged Korean women during the menopausal transition. The strengths of our study include a database from a nationally representative survey. We excluded subjects with a history of diabetes, hypertension, or dyslipidemia because underlying diseases may affect cardiovascular risk factors. Furthermore, we used well-matched age and BMI postmenopausal and premenopausal women, which enabled us to perform group comparisons without bias.

There were several limitations in this study. First, because the study was retrospective and observational, we could not identify a cause-and-effect relationship or mechanism underlying the changes in cardiovascular risk factors during the menopausal transition. Although we adjusted for potential confounding factors (age and BMI), residual confounding factors (smoking, alcohol intake, physical activity, dietary habits, or family history of premature cardiovascular diseases) could have influenced the results of our study. The exclusion of subjects with missing data may have introduced a selection bias. We identified the menopausal status based on self-reports, and therefore, misclassification of menopausal status may have occurred. We did not exclude women on hormone replacement (which could affect the lipid parameters). Although BMI is the standard measure to define obesity, it does not represent body fat composition. We did not evaluate body fat distribution known to be affected by menopausal status.

In conclusion, we observed an association between menopausal status and increased total cholesterol levels only in non-obese subjects among middle-aged Korean women. Therefore, intensive monitoring and treating lipid status, including lifestyle modification education, would be needed to prevent cardiovascular events during the menopausal transition, especially in non-obese subjects.

## Author Contributions

**Conceptualization:** Do Kyeong Song, Young Sun Hong, Yeon-Ah Sung, Hyejin Lee.

**Formal analysis:** Do Kyeong Song, Yeon-Ah Sung.

**Investigation:** Do Kyeong Song.

**Methodology:** Young Sun Hong, Yeon-Ah Sung, Hyejin Lee.

**Project administration:** Young Sun Hong, Yeon-Ah Sung.

**Supervision:** Young Sun Hong, Yeon-Ah Sung, Hyejin Lee.

**Validation:** Do Kyeong Song, Hyejin Lee.

**Writing – original draft:** Do Kyeong Song.

**Writing – review & editing:** Hyejin Lee.

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
