## [Decision Letter · Decision Letter 0]

19 Jan 2023

PONE-D-22-33212The effect of menopause on cardiovascular risk factors according to body mass index in middle-aged Korean womenPLOS ONE

Dear Dr. Lee,

Thank you for submitting your manuscript to PLOS ONE. After careful consideration, we feel that it has merit but does not fully meet PLOS ONE’s publication criteria as it currently stands. Therefore, we invite you to submit a revised version of the manuscript that addresses the points raised during the review process.

We look forward to receiving your revised manuscript.

Kind regards,

Aysha Almas, MBBS, FCPS, MSc

Academic Editor

PLOS ONE

Journal Requirements:

Additional Editor Comments:

1. Abstract : the results do not give actual figures , like mean age is not mentioned, and BMI is not mentioned and cholesterol. Statistical measures are not mentioned

2. Line 53, second paragraph starts with ‘however”. Please change that .

3. Please describe the study outcome clearly

Reviewers' comments:

Reviewer's Responses to Questions

**Comments to the Author**

1. Is the manuscript technically sound, and do the data support the conclusions?

Reviewer #1: Yes

Reviewer #2: Yes

Reviewer #3: Yes

2. Has the statistical analysis been performed appropriately and rigorously? 

Reviewer #1: Yes

Reviewer #2: Yes

Reviewer #3: I Don't Know

3. Have the authors made all data underlying the findings in their manuscript fully available?

Reviewer #1: Yes

Reviewer #2: Yes

Reviewer #3: Yes

4. Is the manuscript presented in an intelligible fashion and written in standard English?

Reviewer #1: Yes

Reviewer #2: Yes

Reviewer #3: Yes

5. Review Comments to the Author

Reviewer #1: This is an interesting study. BMI based difference in cardiovascular risk factors is well knows factor evident from your background too. I wonder if this is due to the limitation of available data that other factors such as smoking status, Family history of premature CVD are not assessed? Post-menopausal women are older than menopausal women are a universal fact. One thing that need some re writing is the result part. It was difficult for me to make a sense of results and i think partly it is due to the fact that there is duplication of what is written in words as it is in table. Can you make your result part easier for the readers? so that they make a good sense.

Reviewer #2: The study investigates an important question to evaluate the effect of menopause on cardiovascular risk factors, accounting for age and BMI. The subjects were analyzed in two groups: premenopausal and postmenopausal. All participants were free from traditional cardiovascular risk factors like diabetes mellitus, dyslipidemia and hypertension. They used a nationally representative sample. The menopausal status showed a significant relationship with increased glucose and cholesterol levels independent for age and BMI, particularly in non-obese women. This highlights the importance of monitoring and addressing the lipid levels to prevent cardiovascular disease during menopause tradition, particularly in the non-obese women ( traditionally may be considered low risk). Menopausal status was an independent predictor predictor of higher HbA1C levels. Statistical analysis was robust using multiple linear regression to determine independent association between menopausal status and total cholesterol after controlling for age , weight status, systolic blood pressure. They also looked at association of menopause and glycemic status (HbA1C) after controlling for age, BMI, systolic blood pressure and total cholesterol.

Reviewer #3: The topic is interesting. The authors have discussed the effect of menopause on cardiovascular risk factors which are important determinants of morbidity and mortality in menopausal women. The The statistical analysis needs to be verified by a statistician. Overall language used is correct but there were a few grammatical errors that need to be corrected. Discussion should be more crisp.

6. PLOS authors have the option to publish the peer review history of their article (what does this mean?). If published, this will include your full peer review and any attached files.

Reviewer #1: **Yes: **Farhala Baloch

Reviewer #2: **Yes: **Azra Rizwan

Reviewer #3: **Yes: **Azra Amerjee

---

## [Author Response · Author response to Decision Letter 0]

23 Feb 2023

Journal Requirements:

→ Thank you for your comment. We have ensured that the manuscript meets PLOS ONE's style requirements.

→ We did not obtain informed consent from individuals because we did not collect data for the study (line 89). 

→ The patient records were anonymous before being released by the KNHANES. This study was approved by the Institutional Review Board of Ewha Medical Center (lines 90-91).

→ Thank you for your valuable suggestion. We have reviewed the Reference list and corrected the format of References 3, 11, and 15. 

Additional Editor Comments:

1. Abstract : the results do not give actual figures , like mean age is not mentioned, and BMI is not mentioned and cholesterol. Statistical measures are not mentioned

→ Thank you for highlighting this point. We have added the mean values to the Abstract. As you suggested, we have included the following information in the Abstract: 

“Differences between groups adjusted for age and BMI were assessed.”

2. Line 53, second paragraph starts with ‘however”. Please change that .

→ Thank you for your feedback. We have removed “however” from the first sentence of the second paragraph in the Introduction.

3. Please describe the study outcome clearly

→ Thanks for your advice. As you suggested, we have included the following information in the Discussion, lines 254-255: 

“In conclusion, we observed an association between menopausal status and increased total cholesterol levels only in non-obese subjects among middle-aged Korean women.” 

Reviewers' comments:

Reviewer's Responses to Questions

Comments to the Author

1. Is the manuscript technically sound, and do the data support the conclusions?

Reviewer #1: Yes

Reviewer #2: Yes

Reviewer #3: Yes

2. Has the statistical analysis been performed appropriately and rigorously?

Reviewer #1: Yes

Reviewer #2: Yes

Reviewer #3: I Don't Know

3. Have the authors made all data underlying the findings in their manuscript fully available?

Reviewer #1: Yes

Reviewer #2: Yes

Reviewer #3: Yes

4. Is the manuscript presented in an intelligible fashion and written in standard English?

Reviewer #1: Yes

Reviewer #2: Yes

Reviewer #3: Yes

5. Review Comments to the Author

Reviewer #1: This is an interesting study. BMI based difference in cardiovascular risk factors is well knows factor evident from your background too. I wonder if this is due to the limitation of available data that other factors such as smoking status, Family history of premature CVD are not assessed? Post-menopausal women are older than menopausal women are a universal fact. 

→ Thanks for your advice. We have included the following details in the Discussion, lines 245-248: “Although we adjusted for potential confounding factors (age and BMI), residual confounding factors (smoking, alcohol intake, physical activity, dietary habits, or family history of premature cardiovascular diseases) could have influenced the results of our study.”

One thing that need some re writing is the result part. It was difficult for me to make a sense of results and i think partly it is due to the fact that there is duplication of what is written in words as it is in table. Can you make your result part easier for the readers? so that they make a good sense.

→ Thanks for your feedback. We have revised the Results (lines 138-144) as follows: 

“The association between menopausal status and total cholesterol according to BMI was similar in the subgroup analysis. Subgroup analysis for 138 postmenopausal and 138 age- and BMI-matched premenopausal women showed that postmenopausal women had higher total cholesterol levels than premenopausal women with marginal significance in total (201 ± 25 mg/dL vs. 196 ± 27 mg/dL, P = 0.091) and non-obese subjects (201 ± 25 mg/dL vs. 194 ± 25 mg/dL, P = 0.060); however, total cholesterol levels did not differ between postmenopausal and premenopausal women in obese subjects (206 ± 23 mg/dL 141 vs. 205 ± 36 mg/dL, P = 0.944).” 

Reviewer #2: The study investigates an important question to evaluate the effect of menopause on cardiovascular risk factors, accounting for age and BMI. The subjects were analyzed in two groups: premenopausal and postmenopausal. All participants were free from traditional cardiovascular risk factors like diabetes mellitus, dyslipidemia and hypertension. They used a nationally representative sample. The menopausal status showed a significant relationship with increased glucose and cholesterol levels independent for age and BMI, particularly in non-obese women. This highlights the importance of monitoring and addressing the lipid levels to prevent cardiovascular disease during menopause tradition, particularly in the non-obese women ( traditionally may be considered low risk). Menopausal status was an independent predictor predictor of higher HbA1C levels. Statistical analysis was robust using multiple linear regression to determine independent association between menopausal status and total cholesterol after controlling for age , weight status, systolic blood pressure. They also looked at association of menopause and glycemic status (HbA1C) after controlling for age, BMI, systolic blood pressure and total cholesterol.

Reviewer #3: The topic is interesting. The authors have discussed the effect of menopause on cardiovascular risk factors which are important determinants of morbidity and mortality in menopausal women. 

The statistical analysis needs to be verified by a statistician. 

→ Thanks for your advice. We consulted a statistician and deleted the results of standardized coefficients in multiple linear regression analyses. 

Overall language used is correct but there were a few grammatical errors that need to be corrected. 

→ Thank you for your feedback. We have proofread the manuscript again for grammatical and typographical errors.

Discussion should be more crisp.

→ Thank you for raising this point. We have revised the Discussion as follows:

 “Menopausal status was associated with increased total cholesterol levels only in non-obese women.” (lines 221-222) 

“We excluded subjects with a history of diabetes, hypertension, or dyslipidemia because underlying diseases may affect cardiovascular risk factors.” (lines 239-241) 

“In conclusion, we observed an association between menopausal status and increased total cholesterol levels only in non-obese subjects among middle-aged Korean women.” (lines 254-255)

After adjustment for age and BMI in non-obese subjects (not obese subjects), postmenopausal women exhibited higher HbA1c and total cholesterol levels than premenopausal women. 

→ Thank you for your comment. We have revised the sentence as follows: 

“After adjustment for age and BMI, postmenopausal women exhibited higher HbA1c and total cholesterol levels than premenopausal women only in non-obese subjects.” (Results, lines 133-134)

Subgroup analysis for 138 postmenopausal and 138 age- and BMI-matched premenopausal women 138 showed that postmenopausal women had higher total cholesterol levels than premenopausal women with marginal significance in total (201 ± 25 mg/dL vs. 196 ± 27 mg/dL, P = 0.091) and non-obese subjects (201 ± 25 mg/dL vs. 194 ± 25 mg/dL, P = 0.060) but not in obese subjects (206 ± 23 mg/dL 141 vs. 205 ± 36 mg/dL, P = 0.944).

→ Thank you for your comment. We have revised the Results as follows:

“Subgroup analysis for 138 postmenopausal and 138 age- and BMI-matched premenopausal women showed that postmenopausal women had higher total cholesterol levels than premenopausal women with marginal significance in total (201 ± 25 mg/dL vs. 196 ± 27 mg/dL, P = 0.091) and non-obese subjects (201 ± 25 mg/dL vs. 194 ± 25 mg/dL, P = 0.060); however, total cholesterol levels did not differ between postmenopausal and premenopausal women in obese subjects (206 ± 23 mg/dL 141 vs. 205 ± 36 mg/dL, P = 0.944).” (lines 138-144)

Menopausal status showed a significant relationship with increased total cholesterol levels even after adjusting for age and BMI in non-obese women, not obese women.

→ Thank you for your valuable suggestion. We have revised the Discussion as follows: 

“Menopausal status showed a significant relationship with increased total cholesterol levels, even after adjusting for age and BMI in non-obese women. However, total cholesterol levels did not differ between postmenopausal and premenopausal subjects in obese women.” (lines 156-160)

The prevalence of metabolic syndrome increased during the menopausal transition in 949 women without diabetes, or the metabolic syndrome at baseline was independent of aging in the Study of Women’s Health Across the Nation.

→ Thank you for the suggestion. We have revised the Discussion as follows:

“The prevalence of metabolic syndrome increased during the menopausal transition in 949 women without diabetes or the metabolic syndrome at baseline was independent of aging in the Study of Women’s Health Across the Nation.” (lines 188-190)

---

## [Editor Report · Decision Letter 1]

8 Mar 2023

The effect of menopause on cardiovascular risk factors according to body mass index in middle-aged Korean women

PONE-D-22-33212R1

Dear Dr. Lee,

We’re pleased to inform you that your manuscript has been judged scientifically suitable for publication and will be formally accepted for publication once it meets all outstanding technical requirements.

Kind regards,

Aysha Almas, MBBS, FCPS, MSc

Academic Editor

PLOS ONE
---

## [Editor Report · Acceptance letter]

15 Mar 2023

PONE-D-22-33212R1 

The effect of menopause on cardiovascular risk factors according to body mass index in middle-aged Korean women 

Dear Dr. Lee:

I'm pleased to inform you that your manuscript has been deemed suitable for publication in PLOS ONE. Congratulations! Your manuscript is now with our production department. 

Kind regards, 

on behalf of

Dr. Aysha Almas 

Academic Editor

PLOS ONE